# Impact of *Bacillus licheniformis*-Fermented Products on Growth and Productivity in Heat-Stressed Laying Ducks

**DOI:** 10.3390/ani14081164

**Published:** 2024-04-12

**Authors:** Rafiq Ahmad, Yu-Hsiang Yu, Felix Shih-Hsiang Hsiao, Hsiu-Wei Liu, Chin-Hui Su, Yeong-Hsiang Cheng

**Affiliations:** 1Department of Biotechnology and Animal Sciences, National Ilan University, Yilan 26047, Taiwan; d1033004@ems.niu.edu.tw (R.A.); yuyh@niu.edu.tw (Y.-H.Y.); shhsiao@niu.edu.tw (F.S.-H.H.); sherlock0731@tmail.ilc.edu.tw (H.-W.L.); 2Eastern Region Branch, Taiwan Livestock Research Institute, Yilan 268020, Taiwan; chsu@mail.tlri.gov.tw

**Keywords:** *Bacillus licheniformis*, Brown Tsaiya duck, egg quality, heat stress, laying performance

## Abstract

**Simple Summary:**

The poultry industry suffers substantial economic losses under heat stress conditions, which detrimentally impact the productivity, physiology, and immune status of the birds. This study focused on the investigation of *Bacillus licheniformis*-fermented product supplementation on the growth and productivity of laying ducks experiencing heat stress. The results showed that *B. licheniformis*-fermented products could be a valuable dietary supplement for enhancing the growth performance and resilience of ducks under heat-stress conditions, possibly offering a safer and more sustainable alternative to antibiotics in poultry production.

**Abstract:**

The purpose of this study was to assess the impact of various concentrations of *Bacillus licheniformis*-fermented products (BLFP) on the growth and productivity of laying ducks (*Anas platyrhynchos*) subjected to heat stress during eight weeks of a feeding trial. A total of 150 one-day-old Brown Tsaiya ducks of both sexes were divided into five groups, with each group having three replicates and 10 ducks each for evaluation of growth performance. The treatment groups received dietary supplements of BLFP at levels of 0.1%, 0.2%, and 0.3%, along with a group receiving flavomycin (F) at 5 ppm, all over a 24-week period. The fermentation process in this study utilized a *B. licheniformis* strain (ATCC 12713) for the production of the spores through solid-state fermentation. The control group was given a basal diet consisting of yellow corn and soybean meal. The results showed that as compared to the flavomycin group, ducks in the 0.3% BLFP group had significantly higher body weights and better feed conversion rates. In addition, during the three weeks, the BLFP group showed higher feed consumption as compared to the control group. The jejunum villi length was significantly increased in the 0.2% BLPF group as compared to the control and flavomycin groups. This study also found that the flavomycin group had a significantly higher egg conversion rate, while the 0.1–0.3% BLFP groups had improved feed intake and the 0.3% group had significantly enhanced egg yolk color. Additionally, the 0.2% BLFP group showed substantial decreases in IL-1β, TNF-α, IL-6, and IL-10 levels in the liver as well as an uptick in the tight junction protein Occludin gene expression in the colon when compared to the control group. Furthermore, the expression of the heat shock protein 70 in the gut upregulated in the 0.1% and 0.2% BLFP groups. In conclusion, these observations demonstrate that dietary supplementation of 0.2% BLFP is an ideal concentration to increase gut morphology, alleviate inflammatory response, and promote gut integrity in heat-stressed laying ducks.

## 1. Introduction

Heat stress is considered an imminent threat to poultry enterprises in many countries as global temperatures rise. The Intergovernmental Panel on Climate Change has documented that the average ambient temperature increased by 1.53 °C between 2006 and 2015, as compared to temperatures from 1850 to 1900 [1]. This ongoing ascent has been projected to result in unusually hot climates, especially in tropical areas. Climate change has a detrimental effect on the food security chain because of its adverse consequences on agricultural products and livestock [2]. Heat stress resulting from high ambient temperatures, particularly in tropical regions of the world, is a major concern in the duck (*Anas platyrhynchos*) industry, as a consequence of heat stress causing poor production performance as well as substantial mortality and morbidity [3]. Several studies have documented that heat-stressed ducks exhibit various physiological adaptations, including endocrine complications, electrolyte deficiencies, and systemic immunological dysfunction. A localized imbalance of anti- and pro-inflammatory mediators in the gastrointestinal tract as well as across the body may be the consequence of reduced intestinal epithelium integrity and an impaired mucosal immune system response under heat stress [4]. Heat shock protein 70 (HSP70), a family of molecular chaperones, has central importance in cells, especially in response to stressors and various cellular processes [5]. HSP70 plays a key role in protein folding, preserving protein integrity, and preventing misfolded protein aggregation [6]. It is a vital part of the cellular stress response, safeguarding proteins during stressors like heat, toxins, and oxidative stress to ensure cell survival [7,8]. HSP70 modulates the immune response by presenting antigens to immune cells and safeguards cells from stress-induced apoptosis to ensure their survival [9].

In conventional poultry (*Gallus gallus domesticus*) industry production, antibiotics have been one of the most frequently employed supplements to enhance feed conversion, growth rate, and bird health, subsequently strengthening productivity and profitability. The involvement of antibiotic growth promoters in the emergence of multi-drug-resistant microorganisms has raised concerns for global public health. Excessive administration of antibiotics at subtherapeutic doses or persistently for an extended period has detrimental impacts on human as well as animal welfare [10]. As a result, European Union banned the application of antibiotics in poultry feed in 2006 [11]. Constraints on the application of antibiotics in feed increase consumer preference and demand for effective alternatives to avoid an abrupt decrease in animal output and economic losses [11].

Probiotics have received significant regard among poultry nutritionists over the last two decades. Probiotics are feed supplements containing live beneficial bacteria that have a favorable impact on the host’s well-being, physiology, or both, which can improve the gastrointestinal structure, aid in the development of immunity against pathogens, and consequently optimize growth performance [12,13,14,15]. It has been documented that *B. licheniformis* has a unique mechanism to counteract oxygen, which inhibits the growth of pathogens, supports intestinal growth and balance, and helps restore bowel functions [16]. The potential of BLFP to improve poultry performance, particularly in the context of heat stress, represents an intriguing avenue of research. While numerous studies have demonstrated its beneficial effects in broiler chickens, its application in Brown Tsaiya ducks under conditions of heat stress remains relatively unexplored. Therefore, this research aims to address this knowledge gap by investigating the impact of BLFP supplementation on Brown Tsaiya ducks subjected to heat stress.

## 2. Materials and Methods

### 2.1. Ethics Statement

Research on animals was conducted according to the institutional committee on animal use (LRIIL IACUC Approval No. 111-06). All experiments were conducted in Wujie (Yilan), Taiwan (latitude 24°46′00″ N and longitude 121°45′00″ E). The experimental period was carried out between 1 June and 30 August 2022.

### 2.2. Experimental Design

A comprehensive study was conducted to investigate the potential of BLFP as a nutritional intervention for mitigating the detrimental effects of heat stress on Brown Tsaiya ducks. During Taiwan’s summer (June to August), this study explored the efficacy of BLFP on growth performance in Brown Tsaiya ducks under heat stress using temperature and humidity records as shown in Figure 1. Before the formal test, the duck houses were cleaned and disinfected in all areas. Over 18 weeks, the ducks received *ad libitum* diet and water, with feed adjusted as required. The feed intake, body weight, and feed conversion ratio were recorded. From day 1 to 4 weeks, the ducks had warming lamps, transitioning to brooding and rearing feed strategies. For immunizations, the ducks were given a subcutaneous poultry cholera vaccine injection. The experiment involved the assessment of various performance parameters, including growth performance, intestinal morphology, egg production, immune response, and the expression of specific proteins.

### 2.3. Animal Subjects

A total of 150 one-day-old Brown Tsaiya ducks of both sexes were divided into five groups, with each group having three replicates and 10 ducks each (five male and five female) (sourced from Yilan Branch of Livestock Experimental Institute, Agriculture Committee of the Executive Yuan) with similar initial body weights. These ducks were selected for their well-known potential for high feed efficiency, making them a relevant model for investigating the impact of BLFP under such conditions.

### 2.4. Experimental Diets

The ducks were randomly assigned to different dietary groups, each with a specific level of BLFP supplementation. The experimental diets were formulated for three distinct phases for the determination of the specific experimental parameters: the rearing period is for the growth performance (0 to 8 weeks of age), the growing period for the measurement of the rectal temperature and respiratory rates (8 to 18 weeks of age), and the laying period for the evaluation of the egg production and egg quality (after 18 weeks of age). During the initial 8 weeks, the ducks were provided with brooding feed, and after that period, rearing feed was administered. At 15 weeks of age, the ducks were categorized into five groups based on their body weight and transferred to individual caged duck houses. The control group was fed a corn-soybean meal basal diet. The experimental *B. licheniformis* were of strain ATCC12713 and spore products for the treatments were generated through solid-state fermentation. These groups included the (1) Control group (CON), receiving a basal diet; (2) Flavomycin (F) group, with 5 ppm Flavomycin supplementation; (3) 0.1% BLFP (0.1%); (4) 0.2% BLFP (0.2%); and (5) 0.3% BLFP (0.3%). For the latter three groups, corresponding amounts of BLFP were incorporated into the feed (equivalent to 1 × 10^8^ CFU spores/kg, 2 × 10^8^ CFU spores/kg, and 3 × 10^8^ CFU spores/kg of feed, respectively). Once the egg production rate reached 5%, the regular feed was replaced with an egg production-specific feed, following a two-week feed treatment period. The composition of these diets, including ingredients and calculated nutritional values, are outlined in Table 1.

### 2.5. Data Collection

Several key parameters were measured to assess the effects of BLFP supplementation on the ducks.

### 2.6. Growth Performance

A total of 150 one-day-old Brown Tsaiya ducks of both sexes were divided into five groups, with each group having three replicates and ten ducks each (five males and five females) with uniform initial body weight. The following parameters were monitored: body weight (BW), feed intake (FI), and feed conversion rate (FCR). These measurements were taken at weekly intervals from 1 to 6 weeks of age, providing insights into the ducks’ growth trajectories and feed efficiency.

### 2.7. Rectal Temperatures and Respiratory Rate

A total of 90 eight-week-old Brown Tsaiya ducks of both sexes were divided into five groups, each group having three replicates, and six ducks per replicate (three males and three females). Rectal temperatures and respiratory rates were recorded at the age of 8~18 weeks from July to August, specifically in the midafternoon around 1 pm experiencing heat stress conditions. A handheld digital thermometer (Testo thermometer, Taichung, Taiwan) with an accuracy of ±0.1 °C was employed for rectal temperature measurement. The respiratory rate of each duck was meticulously recorded within the initial minute following stabilization. Three randomly selected males and three females from each cage were measured and gender differences in respiratory rates were also investigated.

### 2.8. Intestinal Morphology

To assess how BLFP affects intestinal health, 90 eight-week-old Brown Tsaiya ducks were divided into five treatments, each treatment having three replicates and six ducks per replicate (three males and three females). At the end of the 18th week, intestinal samples were collected to investigate morphological alterations in the duodenum, jejunum, and ileum. To evaluate the structural integrity of the intestinal mucosa, variables including villus height, crypt depth, and the villus-to-crypt ratio were assessed. Each intestinal segment (duodenum, jejunum, and ileum) was cut into three cross-sections, each of which was stained with hematoxylin and eosin after being embedded in paraffin. We collected 15 well-oriented crypt-villus units in triplicate for each intestinal cross-section sample. Villus height and crypt depth were measured in micrometers (µm) using an image processing and analysis system (version 6.0, Image-Pro Plus). The villus height extended from the villus tip to the villus-crypt junction, while crypt depth was determined by the depth of invagination between neighboring villi.

### 2.9. Egg Production and Egg Quality

A total of 90 eighteen-week-old Brown Tsaiya ducks were allocated into five groups with treatment consisting of three replicates and six ducks with each replicate. In the laying period (after 18 weeks of age), parameters including feed intake (FI), body weight (BW), egg production rate, egg weight (EW), average daily feed intake (egg mass), egg dimensions, eggshell quality, and yolk color were evaluated to determine the influence of BLFP on egg production and quality. After 22 weeks, we randomly selected 18 eggs from each group for in-depth quality analysis, measuring albumen height, Haugh units, eggshell thickness, eggshell strength (measured with a digital egg tester after eggs were weighed and cracked open), and yolk color, which was gauged using a Roche colorimeter (Yolk color fan, Heerlen, The Netherland). This comprehensive evaluation provided insights into BLFP’s influence on Brown Tsaiya duck performance during the laying phase.

### 2.10. Quantitative Polymerase Chain Reaction and Western Blotting

Quantitative real-time polymerase chain reaction (qRT-PCR) tests were applied to determine the gene expressions of pro-inflammatory cytokines IL-1β, IL-6, and TNF-α, as well as the anti-inflammatory cytokine IL-10, in the liver to investigate the ducks’ immune responses to BLFP supplementation. For this purpose, 90 eight-week-old ducks were divided into five treatments, each group having three replicates and six ducks per replicate. Similarly, the expression levels of HSP70 and tight junction proteins (Occludin and Claudin- 1) were examined to gain insights into the potential mechanisms underlying the effects of BLFP on gut health and integrity. For qRT-PCR assays, the total RNA was proficiently isolated by employing the TRIzol reagent (Sigma Aldrich, St. Louis, MO, USA) in strict accordance with the provided manufacturer’s guidelines. To enable reverse transcription, the Primer probe-SYBR Green (BIO-RAD, Hercules, CA, USA) was adeptly employed. Subsequently, the quantitative real-time PCR test was carried out using the Real-Time PCR Detection System (BIO-RAD, CFX Optic Module, Singapore) along with the iTaqTM Universal SYBR Green Supermix (BIO-RAD, Hercules, CA, USA). Normalization of the gene expression data was conducted, referencing the internal control β-actin (Table 2). This was achieved through a refined methodology that involved the comparison of Ct values using the formula 2^−ΔΔCt^. Total protein content was extracted from the liver using a lysis buffer containing RIPA (Beyotime, Beijing, China). Protein concentrations were determined using an improved BCA protein assay kit (Beyotime). Equal amounts of protein were electrophorized in a 12% sodium dodecyl sulfate-polyacrylamide gel. Proteins were transferred to polyvinylidene difluoride (PVDF) membranes. The membranes were blocked for 2 h at 37 °C with 5% bovine serum albumin (BSA) dissolved in PBST (containing 0.5% Tween-20). The membranes were treated with the HSP70 primary antibody (Rabbit antibody, Cell Signaling Technology, Danvers, MA, USA) overnight at 4 °C following three PBST washes. This procedure was performed three times with fresh PBST. The membrane was incubated with the secondary antibody (Rat antibody, Cell Signaling Technology, Danvers, MA, USA) for 1 h at 37 °C, washed three times with PBST, and then fluorescence color analysis was performed using UVP cold light imaging equipment (Analytikjena, Upland, CA, USA).

### 2.11. Statistical Analysis

Statistical analysis was conducted using the Statistical Analysis System 9.4 software to determine the significance of differences between experimental groups. Analysis of variance (ANOVA) was performed with the General Linear Model procedure (GLM) followed by Tukey’s test to assess the data to compare the significant differences between treatments, each test value is presented as mean ± standard deviation (Mean ± SD). *p* < 0.05 means a significant difference, and *p* < 0.01 means a highly significant difference.

## 3. Results

### 3.1. Growth Performance

The effects of BLFP supplementation on growth performance, body weight, feed intake, and FCR in Brown Tsaiya ducks are shown in Table 3. The body weight of the BLFP group with supplementation of 0.3% was significantly higher as compared to the flavomycin group at the age of six weeks. No statistical differences were found in feed intake between groups. In terms of feed conversion rate, the 0.3% BLFP group had a significantly better feed conversion rate at six weeks of age as compared to the flavomycin group.

### 3.2. Rectal Temperatures and Respiratory Rate

The study investigated rectal temperatures and respiratory rates as indicators of the ducks’ responses to heat stress. Rectal temperatures and respiratory rates of heat-stressed Brown Tsaiya ducks are more directly related to the external environment, and diet has a significant effect on rectal temperatures, as shown in Table 4. There is a significant difference between the male duck and the female duck in regards to the respiration rate. The respiration rate of the female duck is significantly higher than that of the male duck. This is a significant challenge for laying eggs because the rapid breathing movement will cause a large amount of carbon dioxide to be exhaled from the body, consequently affecting the thickness and strength of eggshells.

### 3.3. Intestinal Morphology

In the duodenum, the ratio of the villus to crypts in the 0.2% BLFP group tended to be higher than that in the 0.1% and 0.3% BLFP groups. In the jejunum, the villus height in the 0.2% BLFP group was significantly higher than the other groups. Additionally, the villus to crypt ratio also tended to be higher as compared to other groups. In the ileum, the length of the villus in the 0.1% BLFP group was significantly higher among the BLFP groups as well as compared to the control and flavomycin groups, as shown in Table 5.

### 3.4. Egg Production and Egg Quality

It is evident from the results shown in Table 6 that dietary supplementation of 0.1–0.3% BLFP boosted feed intake as compared to the flavomycin group, and 0.3% BLFP enhanced the egg yolk colour of the laying ducks as compared to the other groups. Additionally, the flavomycin group exhibited a significantly higher egg conversion rate.

### 3.5. Pro-Inflammatory Cytokines

The gene expressions of pro-inflammatory cytokines IL-1β and IL-6 in the liver were significantly lower in the flavomycin and 0.2% BLFP groups as compared to the control group, as documented in Figure 2A,C. Similarly, the expression levels in the flavomycin, 0.1% BLFP, and 0.2% BLFP groups of TNF-α were significantly lower than that of the control group. The expression levels of the anti-inflammatory cytokine IL-10 in 0.2% and 0.3% BLFP groups were significantly lower than the control group.

### 3.6. HSP70 and Tight Junction-Associated Gene Expression

The expression level of HSP70 in the liver of the flavomycin group and 0.1% BLFP group was significantly higher than that of other groups as shown in Figure 3A. The gene expression of tight junction protein in the intestine and the expression of the Occludin gene in the 0.1% and 0.2% BLFP groups were significantly higher as compared to the control and 0.3% BLFP groups, as shown in Figure 3B. The expression level of Claudin-1 in the flavomycin group was significantly higher than other groups. The protein content of HSP70 in the intestine in the 0.1% and 0.2% BLFP groups was significantly higher as compared to the flavomycin group, as reported in Figure 3C.

## 4. Discussion

Probiotics are being investigated as a safe and effective substitute for antibiotics as numerous studies have demonstrated their ability to safeguard against intestinal infectious illnesses, increase chicken production efficiency, and improve the quality of chicken products [17]. The introduction of probiotics in poultry diets effectively promotes the proliferation of beneficial microbes as well as preserves an improved intestinal tract, resulting in enhanced broiler growth performance. *B. licheniformis* has additionally captured public attention with the surge in probiotic research [18]. Prior research has demonstrated that adding *B. licheniformis* to drinking water effectively increased broiler growth performance (BW, ADG, and FCR) [19]. Another study revealed that supplementing *B. licheniformis* with a broiler’s diet could boost their BW and ADG [20]. According to Li et al. [21], after supplementation of 0.4% and 0.8% *B. licheniformis* to Pekin ducks’ diet at the ages of 1 and 6 weeks, their body weight and feed consumption were significantly higher as compared to the control group. Deng et al. [22] further demonstrated that supplementation of *B. licheniformis* could improve egg production and feed intake in laying hens under heat stress. Our previous studies have reported that BLFP supplementation increased growth performance of broilers either in challenged or unchallenged models [23,24]. However, the current study revealed that dietary BLFP supplementation was unable to improve growth performance, egg production, and egg quality compared to the control group. This discrepancy may be due to differences in *B. licheniformis* spores between these studies. A previous study has indicated that the effective concentration for *B. licheniformis* spores should be equal to 10^7^ CFU/g of feed for increasing egg production and feed intake in laying hens under heat stress [22]. Similarly, previous studies also found that *B. licheniformis* spore concentration in BLFP should be equal to or greater than 10^7^ CFU/g of feed for improving the growth performance of broilers [23,24]. In the current study, the *B. licheniformis* spore concentration in the diet of the ducks was much lower than our previous studies (10^5^ CFU/g of feed versus 10^7^ CFU/g of feed) [23,24]. Therefore, the ideal concentration of *B. licheniformis* spore in BLFP for improving growth performance, egg production, and egg quality in heat-stressed laying ducks still needs to be investigated in the future.

The intestinal mucosa structure can be used to anticipate the functionality of the digestive system. Higher villus height demonstrates a large surface area for nutrition absorption, while a deeper crypt reveals rapid tissue turnover and intensive preference for new tissue. Impaired gut epithelium integrity could raise the absorption and likelihood of potentially detrimental substances [22]. Additionally, deeper crypts may signify an expansion of the enterocyte proliferative pool, which may be required to facilitate quick epithelium turnover in response to pathogen- or toxin-induced inflammation. According to Ma et al. [25], heat stress damaged intestinal villi and increased crypt depth. Furthermore, Deng et al. [22] documented that heat stress can shorten intestinal villi, deepen crypts, and reduce the ratio of villi to crypt bodies, and *B. licheniformis* supplementation can significantly ameliorate this detrimental condition and preserve the integrity of the intestinal architecture by lowering the expression of pro-inflammatory cytokines. In another study, Lei et al. [16] revealed that their *B. licheniformis*-treated group showed improved gut integrity as compared to the control group. Our previous studies have demonstrated that BLFP supplementation ameliorates gut morphology by increasing villus height in the small intestine of broilers either in challenged or unchallenged models [24,26]. The current study has shown that higher villus height indicators in the jejunum of the 0.2% BLFP group and ileum of the 0.1% BLFP group were observed. This is consistent with previous studies that found supplementation of broiler diets with *B. licheniformis* or BLFP could improve their gut morphology better than control groups [16,22,24,26]. It has been reported that *B. licheniformis* is able to inhibit the growth of common enteric pathogens of poultry by production of antimicrobial lipopeptides and modulate gut microbial composition by increasing the abundance of the probiotics in broilers [23,24]. Therefore, the beneficial effects of BLFP on the gut morphology of heat-stressed laying ducks may be attributed to decreased toxicity of enteric pathogens and more effective villi protection by increased intestinal probiotics.

We demonstrated that the serum IL-1β and IL-6, as well as IL-10 in the liver, were elevated during heat treatment but were significantly reduced in the BLFP-supplemented group. Wang et al. [27] reported that gene expression of IL-6 and IL-10 in the liver was significantly increased, while *B. subtilis* dietary inclusion significantly reduces the expression levels of IL-6 and IL-10 under heat stress treatment, which is consistent with the results of the current experiment. Deng et al. [22] showed that under heat stress, the serum TNF-α level increased, and administration of *B. licheniformis* could reduce it, having no significant difference with the normal temperature control group. A previous study demonstrated that BLFP supplementation could modulate cecal microbial composition by decreasing the abundance of potential pathogens (Gram-positive and Gram-negative bacteria) in broilers [28]. This implies that dietary inclusion of BLFP may reduce pathogen-derived endotoxins or virulence factor invasion in the blood of heat-stressed laying ducks via modulation of gut microbiota, thereby preventing systemic inflammation and lowering stress response. Whether *B. licheniformis*-produced metabolites in the gut can regulate along the hypothalamic-pituitary-adrenal axis to alleviate stress responses and subsequent inflammatory stimulation still needs to be elucidated.

It has been reported that heat stress increases the expression of HSP70 in broilers and laying hens, and that the degree of heat stress has a positive correlation with the expression of HSP70 [2]. HSP70 functions as a molecular chaperone to facilitate appropriate protein folding and prevent protein aggregation in response to heat stress in poultry [29]. HSP70 protects thermosensitive proteins from denaturation by stabilizing them and promotes the transport of misfolded proteins to cellular compartments for refolding or destruction, preserving cellular stability [29]. A previous study indicated that HSP70 mRNA expression escalated in a time-dependent manner from 0 to 4 h of heat exposure and thereafter exhibited a gradual decreasing pattern over an 8 h recovery period [30]. In the current study, the HSP70 expression level in the liver for the 0.2% BLFP group was significantly lower as compared to the flavomycin and 0.1% BLFP groups. Furthermore, the HSP70 expression level in the intestine for the 0.2% BLFP group was higher as compared to the other groups, although there was no statistical difference. According to a previous study, the HSP70 mRNA will be transcribed first in response to heat stress and then return to normal levels in chicken peripheral blood mononuclear cells [30]. Thereafter, the HSP70 mRNA level declined and HSP70 protein levels will be increased to promote the appropriate protein folding during heat stress by translation [29,30]. Hence, increased HSP70 protein levels in the intestine of 0.2% BLFP group implies that BLFP supplementation may promote the organs to become more sensitive and responsive to heat stress in laying ducks. In addition, HSP70 can strengthen and control intestinal tight junctions, making the intestinal barrier more consolidated [31]. In cell culture studies, the HSP70 protein was overexpressed and considerably enhanced the integrity of epithelium [31,32]. In this study, 0.2% BLFP supplementation effectively upregulated Occludin mRNA expression, demonstrating the beneficial effect of BLFP on the intestinal barrier of laying ducks. This is consistent with previous studies that found that *B. licheniformis CGMCC 1.3448* optimizes epithelial barrier integrity in laying hens [33] and *B. licheniformis B26* in broiler chickens [34], confirmed by increased mRNA expression of ZO-1, Occludin, and Claudin-1 in the intestines. It has been demonstrated that preadministration of *B. licheniformis* can reduce intestinal damage and enhance gut barrier function during, as well as after, the onset of heat stroke [35]. In addition, our findings have shown that the 0.2% BLFP-supplemented group had significantly higher villus height in the jejunum compared to the other groups. Taken together, the increased HSP70 protein levels in the intestine of the 0.2% BLFP group indicates a potential role of BLFP in heat stress adaptation. Furthermore, the upregulation of tight junction proteins, particularly Occludin mRNA levels in the 0.2% BLFP group, suggests an enhancement of intestinal barrier function. This may lead to reduced intestinal permeability and better protection against pathogen invasion and subsequent inflammatory response.

## 5. Conclusions

In conclusion, 0.2% BLFP supplementation showed the best results, characterized by an increase in gut morphology and integrity and a lower inflammatory response in heat-stressed laying ducks. These findings hold practical significance for the poultry industry, suggesting that BLFP stands as a promising nutritional intervention to improve duck farming practices.

## Figures and Tables

**Figure 1 animals-14-01164-f001:**
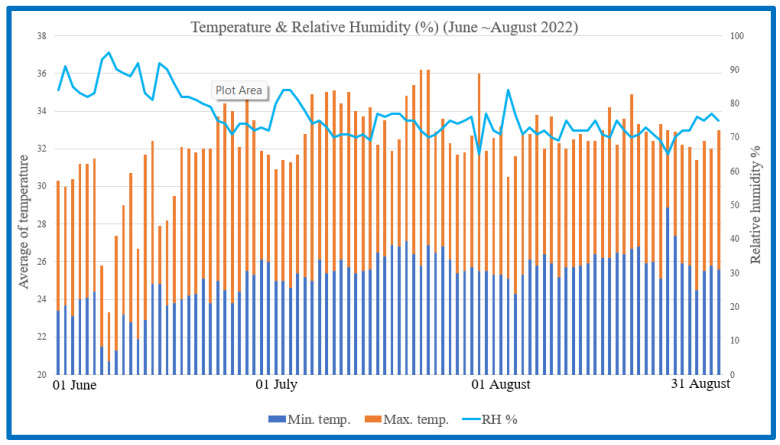
Maximum, minimum temperature, and relative humidity from June to August 2022.

**Figure 2 animals-14-01164-f002:**
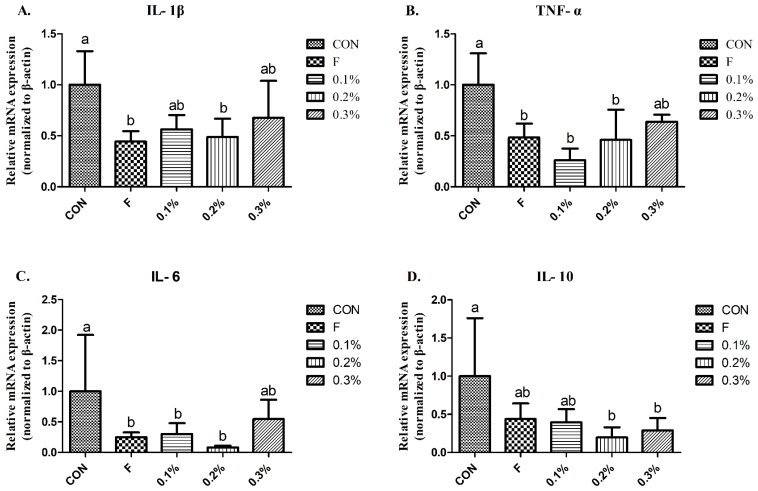
Effect of inflammation-associated gene expression in liver of Brown Tsaiya ducks (n = 3). ^a,b^ Means of a row with no common superscript are significantly different (*p* < 0.05). Values are presented as mean ± SD. (**A**) IL- 1 β, (**B**) TNF- α, (**C**) IL-6, (**D**) IL-10.

**Figure 3 animals-14-01164-f003:**
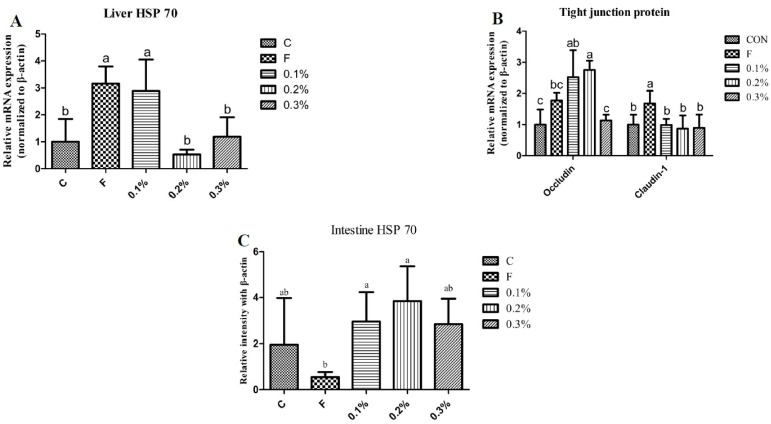
Effect of heat stress on HSP70, tight junction mRNA intestine, and HSP70 western blot of Brown Tsaiya ducks (n = 3). ^a–c^ Means of a row with no common superscript are significantly different (*p* < 0.05). Values are presented as mean ± SD. (**A**) Liver HSP70 (**B**) Tight junction protein (**C**) Intestine HSP70.

**Table 1 animals-14-01164-t001:** Diet composition in the rearing period (Starter diet: 0 to 8 weeks of age), growing period (Grower diet: 8 to 18 weeks of age), and laying period (Layer diet: after 18 weeks of age).

Ingredients	Starter Diet	Grower Diet	Layer Diet
Yellow corn	55.54	51.94	49.93
Soybean meal (44%)	25.30	10.00	27.00
Wheat flour middling	10.30	20.00	-
Wheat bran	-	10.00	6.50
Fish meal	2.00	-	3.30
Yeast powder	3.00	2.00	2.00
Rice hull powder	-	2.40	-
Soybean oil	1.10	-	2.50
Limestone	1.10	1.60	6.60
Di-calcium phosphate	1.10	1.50	1.50
Salt iodized	0.30	0.30	0.40
Choline chloride (50%)	0.08	0.08	0.08
Lysine-HCl	-	-	0.01
DL-methionine	0.05	0.05	0.05
Vitamin premix ^a^	0.03	0.03	0.03
Mineral premix ^b^	0.10	0.10	0.10
	**Calculated Value**		
ME (kcal/kg)	2900	2660	2700
Crude protein (%)	19.5	13.5	20.0
Calcium (%)	0.81	0.94	3.05
Available phosphorus (%)	0.36	0.44	0.44
Methionine (%)	0.38	0.27	0.39
Lysine (%)	1.05	0.60	1.11

**^a^ Supplied per kg diet**: niacin 60 mg, calcium pantothenate 18 mg, vitamin E 22.5 mg, vitamin B2 9 mg, vitamin B6 6 mg, vitamin K3 6 mg, vitamin B1 3 mg, folic acid 1.5 mg, vitamin B12 0.03 mg, biotin 0.03 mg, vitamin A 15,000 IU, vitamin D3 3000 IU. **^b^ Supplied per kg diet:** CuSO_4_•5H_2_O 8 mg, FeSO_4_ 120 mg, ZnSO_4_•H_2_O 60 mg, Mn_3_O_4_ 60 mg, KIO_3_ 0.7 mg, Na_2_SeO_3_ 0.2 mg, CaCO_3_ 0.2 mg.

**Table 2 animals-14-01164-t002:** Gene-specific sequence primers used in real-time quantitative PCR.

Gene	Accession No.		Primer Sequence (5′-3′)	Product Size (bp)
β-actin	NM_001310421.1	Forward:	TGATATTGCTGCGCTCGTTGT	193
		Reverse:	CAGGGTCAGGATACCTCTTTTGC	
HSP70	GI281323015	Forward:	CCCCCAGATCGAGGTTACTTT	200
		Reverse:	CTCCCACCCGATCTCTGTTG	
IL-1β	DQ393268	Forward:	TCGACATCAACCAGAAGTGC	185
		Reverse:	GAGCTTGTAGCCCTTGATGC	
TNF-α	XM_005027491.3	Forward:	ACCCCGTTACAGTTCAGACG	140
		Reverse:	TAGCCATGTCAATGCTCCTG	
IL-6	XM_027450925	Forward:	TTCGACGAGGAGAAATGCTT	150
		Reverse:	CCTTATCGTCGTTGCCAGAT	
IL-10	NM_001310368.1	Forward:	GGGAGAGGAAACTGAGAGATGT	112
		Reverse:	TCCTTTCCTCTTAGTCCAGCTC	
IFN-γ	AJ012254	Forward:	GCTGATGGCAATCCTGTTTT	247
		Reverse:	GGATTTTCAAGCCAGTCAGC	
IL-2	AF294323	Forward:	GCCAAGAGCTGACCAACTTC	137
		Reverse:	ATCGCCCACACTAAGAGCAT	
IL-4	XM_038186630.1	Forward:	GGCAATGAGGTAAGACGGGA	232
		Reverse:	AGCGTTTTGTGCCCATGGAT	
IL-5	XM_046927093.1	Forward:	CACATCAGGACCATGAGGACC	239
		Reverse:	CCGAATCTCCTCATCTCGGG	
Claudin-1	XM 013108556.1	Forward:	TCATGGTATGGCAACAGAGTGG	179
		Reverse:	CGGGTGGGTGGATAGAAG	
Occludin	XM 013109403.1	Forward:	CAGGATGTGGCAGAGGAATACAA	160
		Reverse:	CCTTGTCGTAGTCGCTCACCAT	

**Table 3 animals-14-01164-t003:** Effects of *B. licheniformis*-fermented products on growth performance in Brown Tsaiya ducks (1~6 weeks).

			BLFP				
Items	Control	Flavomycin	0.1%	0.2%	0.3%	SEM	*p*-Value
Body weight (g)							
3 wk	272.09	279.00	323.64	285.40	303.08	10.95	0.58
6 wk	838.51 ^ab^	741.3 ^b^	882.26 ^ab^	822.99 ^ab^	891.31 ^a^	17.07	0.05
Feed intake (g)							
3 wk	884.08	948.70	942.50	923.02	907.90	18.96	0.89
6 wk	3617.50	3167.00	3277.00	3093.00	3259.00	83.92	0.38
1~6 wk	4501.58	4115.70	4219.50	4016.02	4166.90	88.35	0.59
Feed conversion ratio (G/F)							
3 wk	4.35	4.21	3.26	3.59	3.51	0.22	0.47
6 wk	6.41	7.07	6.05	5.95	5.60	0.16	0.06
1~6 wk	5.63 ^ab^	5.84 ^a^	4.99 ^ab^	5.13 ^ab^	4.82 ^b^	0.12	0.03

^a,b^ Means of a row with no common superscript are significantly different (*p* < 0.05) (n = 3).

**Table 4 animals-14-01164-t004:** Effect of *B. licheniformis*-fermented products on rectal temperatures and respiratory rate in heat-stressed Brown Tsaiya ducks (8~18 weeks).

Parameters	Control		Flavomycin		0.1%		0.2%		0.3%		*p*			
	Male	Female	Male	Female	Male	Female	Male	Female	Male	Female	SEM	Diet	Gender	Diet × Gender
Rectal temperature	42.29	42.03	42.47	42.34	42.26	42.27	42.18	42.48	42.34	42.22	0.03	0.01	0.54	0.09
Breath count	22.19	28.75	24.97	32.89	25.06	29.94	25.31	28.22	25.22	34.86	0.54	0.3363	<0.0001	0.5898

Means of a row with no common superscript are significantly different (*p* < 0.05) (n = 3).

**Table 5 animals-14-01164-t005:** Effect of *B. licheniformis*-fermented products on intestinal morphology in heat-stressed Brown Tsaiya ducks (18 weeks).

				BLPF				
Items		Control	Flavomycin	0.1%	0.2%	0.3%	SEM	*p*-Value
Duodenum								
	Villus height (μm)	811.89	812.02	701.59	776.62	709.78	29.39	0.63
	Crypt depth (μm)	21.99	21.28	28.11	24.01	31.62	1.43	0.11
	Villus: Crypt	39.84	39.09	28.80	34.66	25.87	1.94	0.08
Jejunum								
	Villus height (μm)	695.69 ^b^	752.44 ^b^	825.19 ^b^	995.68 ^a^	804.33 ^b^	23.32	<0.01
	Crypt depth (μm)	33.86	24.88	29.31	29.11	27.59	1.29	0.28
	Villus: Crypt	23.08	31.74	29.39	37.74	31.74	1.64	0.07
Ileum								
	Villus height (μm)	627.1 ^bc^	596.78 ^c^	763.57 ^a^	731.68 ^ab^	753.69 ^ab^	21.42	0.03
	Crypt depth (μm)	42.74	40.59	48.43	43.29	37.20	1.98	0.50
	Villus: Crypt	16.33	14.88	19.00	17.57	24.28	1.40	0.25

^a–c^ Means of a row with no common superscript are significantly different (*p* < 0.05) (n = 3).

**Table 6 animals-14-01164-t006:** Effects of *B. licheniformis*-fermented products on egg production and egg quality in Brown Tsaiya ducks (22~28 weeks).

			BLFP				
	Control	Flavomycin	0.1%	0.2%	0.3%	SEM	*p*-Value
Feed intake (g)	161.1 ^a^	139.1 ^b^	171.0 ^a^	164.4 ^a^	167.1 ^a^	2.94	<0.0001
Body weight (g)	1263.3 ^ab^	1200.9 ^b^	1244.9 ^ab^	1197.6 ^b^	1340.7 ^a^	16.08	0.043
Production rate (%)	88.9	90.2	92	89	89	0.01	0.7291
Egg Weight (g)	64.3 ^a^	60.2 ^ab^	61.9 ^ab^	62.5 ^a^	60.1 ^b^	0.54	0.0301
Feed conversion ratio	2.5 ^c^	2.3 ^d^	2.8 ^a^	2.6 ^bc^	2.8 ^ab^	0.02	0.0027
Egg width (mm)	42.9	42.7	43	43.3	40.6	0.58	0.2929
Egg height (mm)	59.1	57.7	58.9	59.1	57.9	0.26	0.1602
Shape index	72.64	74	73.2	73.3	70.6	1.01	0.6872
Break strength (N)	48.69	52.3	47.1	50.9	54.6	1.20	0.1791
Albumin height	8.2	7	8.2	6.9	7.8	0.23	0.0646
Haugh	88	82.5	89.1	80.7	85.6	1.60	0.1959
Yolk color	12.1 ^b^	12.1 ^b^	12 ^b^	11.8 ^b^	12.4 ^a^	0.06	0.0217
Eggshell thickness (mm)							
Top	0.412	0.372	0.368	0.369	0.386	0.0043	0.1293
Middle	0.400	0.408	0.387	0.416	0.400	0.0027	0.3629
Bottom	0.382	0.374	0.357	0.38	0.405	0.0038	0.0964

^a–d^ Means of a row with no common superscript are significantly different (*p* < 0.05) (n = 3).

## Data Availability

The data supporting the results of this article will be available by the authors on request.

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
