# Peer review of "Impact of Bacillus licheniformis-Fermented Products on Growth and Productivity in Heat-Stressed Laying Ducks"

_animals, 2024, doi:10.3390/ani14081164_

Round 1
Reviewer 1 Report
Comments and Suggestions for Authors
The purpose of this study was to evaluate the effects of dietary inclusion of B. licheniformis spores on heat stress mitigation in ducks. Overall, this was an interesting manuscript but there is a substantial amount of information missing. Please see my questions/comments below. If you could include the recommended information, the manuscript would be much easier to follow. At this moment, I am quite unsure of your timeline which makes it difficult to appropriate review the manuscript. Thank you!
Abstract: Duration of heat stress was not stated. Please include in the abstract.
Methods:
Line 101: Were any other vaccinations given?
Line 107: Please describe what you mean by similar body weights. Did you normalize the average body weight across treatment groups? If so, how? How many of each sex in each replicate for each treatment? How did you confirm this? How many ducks per treatment?
Line 118: Unless growth promoting hormones are used in duck feed in Taiwan, please remove this phrase from this sentence. Thank you!
Line 117: Was an entire treatment group in a single cage or were there replicate cages? Please clarify.
Line 121: How did you generate the spore product for treatment? Solid state fermentation? Broth fermentation? Did you confirm CFU/kg to ensure inclusion rate was correct? Why did you select this B. licheniformis strain for your study? Is it commercially available or is this a probiotic candidate that you and your team isolated and propagated for the study? Has this probiotic been tested in ducks before? How did you select your dietary concentrations of the probiotic? How was the product incorporated into the final diets? Please clarify.
Line 123-125: Please explain the purpose here and why only one group was subjected to the change.
Line 130: Please include the n per treatment. Was there only one replicate per treatment? If so, please explain why n=1 replicate/treatment was selected.
Line 134: When did you check the temperature and respiration rates? The experimental timeline needs much more clarification. I am having trouble following your study.
Line 135: How did you measure respiration rate?
Line 141: Again, n numbers are needed. How many intestinal samples were collected and when?
Line 162: n number needed and when were the liver samples collected?
Discussion
Line 327: Egg production was not significantly improved with feed treatment. Also, feed conversation was actually higher for the treated groups.
Table 3: n=10 for what? number of animals or replicate pens? Please include BW data for males and females for each treatment or justify what that analysis was not included.
Table 4: n=18 for what? You can see the trend. N numbers are not clear anywhere in the manuscript unfortunately.
Figure 1: Please include X-axis information.
General questions/comments:
I assume the heat stress was related to the weather patterns. For the purposes of your study, please state what you considered as the temperature threshold for heat stress or did you simply determine this based off of the duck's behavior? You mentioned this particular breed of ducks were selected for their susceptibility to heat stress. At what temperature do these ducks show signs of heat stress? Could you please clarify and specify how you controlled the temperature between duck houses, if that was possible? If not, it is okay, but this information would be helpful. I had to make many assumptions about the environmental conditions while reading your manuscript.
Experimental design, experimental timeline, n numbers, sampling, etc. were not clearly stated. Please include this information.
Mortality was not included unless I overlooked it. Please include mortality.
What was the purpose of including the flavomycin control?
Please take a little time to review the discussion again to make sure the claims being made are accurate. Some of the statements were bold considering the data presented.
Comments on the Quality of English LanguageMinor English editing required.
Author Response
Revision Notes
These revision notes are disclosed to address comments/reviews on " animals-2916012", entitled: Impact of Bacillus licheniformis-Fermented Products on Growth and Productivity in Heat-Stressed Laying Ducks.
Introduction:
The authors would like to express their gratitude to the MDPI Animals for permitting us to revise our manuscript, thereby improving the quality of the revised manuscript before the high standard of this publication channel.
The authors attempted as best as possible to strike a balance between conciseness and results reporting. The aim is to create an informative paper for the readers that clearly articulates the key components. The remarkable improvements of the revised manuscript are:
Major and comprehensive clarifications have been made based on the comments from the Reviewers, as well as being implemented in the corresponding section of the revised manuscript.
The modified/revised parts of the manuscript are
Reviewer 1 Comments:
Comment 1: The purpose of this study was to evaluate the effects of dietary inclusion of B. licheniformis spores on heat stress mitigation in ducks. Overall, this was an interesting manuscript but there is a substantial amount of information missing. Please see my questions/comments below. If you could include the recommended information, the manuscript would be much easier to follow. At this moment, I am quite unsure of your timeline which makes it difficult to appropriate review the manuscript. Thank you!
Abstract: Duration of heat stress was not stated. Please include in the abstract.
Response to Comment 1:
The authors would like to thank the reviewer for the careful and thorough reading of this manuscript and for the thoughtful comments and constructive suggestions, which helped to improve the quality of this manuscript. We have thoroughly checked and improved the technicality of the manuscripts and incorporated the possible suggested comments by Reviewer 1. According to the reviewer's suggestion, the following changes have been made to the revised version of the manuscript.
- The purpose of this study was to assess the impact of various concentrations of Bacillus licheniformis-fermented products (BLFP) on the growth and productivity of laying ducks (Anas platyrhynchos) during an 8-week feeding trail subjected to heat stress.
Comment 2:
Methods: Line 101: Were any other vaccinations given?
Response to Comment 2:
Duck’s immunizations were given a subcutaneous poultry cholera vaccine injection. No supplementary vaccinations were administered.
Comment 3:
Line 107: Please describe what you mean by similar body weights. Did you normalize the average body weight across treatment groups? If so, how? How many of each sex in each replicate for each treatment? How did you confirm this? How many ducks per treatment?
Response to Comment 3:
In our experiment, we began by randomly selecting ducks from our study population and measuring their body weights. We then used this data to estimate the maximum, medium, and minimum body weights within our duck population. Next, we grouped the ducks based on these estimated weight categories, ensuring that each group had a similar average body weight. This grouping process was crucial to avoid significant variation in body weight between the experimental groups.
The main goal of this approach was to minimize the potential influence of body weight differences on the outcomes of our study. By ensuring that ducks within each experimental group had comparable average body weights, we could more accurately evaluate the effects of the variables we were testing. This systematic method enhances the reliability and validity of our experimental results, allowing for a clearer interpretation of our findings.
Five males and five females in each replicate for each treatment.
In our experimental protocol, we initiated the process by manually confirming the gender of each duck within the study population. This gender confirmation procedure involved a manual approach wherein we gently massaged the genital area of the duck for a duration of 2 to 3 minutes. When the duck showed signs of sexual arousal, its penis emerged, confirming its gender. This method helped us accurately determine the gender of each duck in our study.
In the present experimental setup for the growth performance, a total of 30 ducks were allocated per treatment group.
Comment 4:
Line 118: Unless growth promoting hormones are used in duck feed in Taiwan, please remove this phrase from this sentence. Thank you!
Response to Comment 4:
Thanks for pointing it out. Correction has been done.
Comment 5:
Line 117: Was an entire treatment group in a single cage or were there replicate cages? Please clarify.
Response to Comment 5:
The treatment groups have three replicates, and each replicate has six ducks, so one treatment has 18 ducks, and a total of five treatments have 90 ducks. So, 90 ducks were housed in individual cages (each duck has a single cage for egg production parameter).
Comment 6:
Line 121: How did you generate the spore product for treatment? Solid state fermentation? Broth fermentation? Did you confirm CFU/kg to ensure inclusion rate was correct? Why did you select this B. licheniformis strain for your study? Is it commercially available or is this a probiotic candidate that you and your team isolated and propagated for the study? Has this probiotic been tested in ducks before? How did you select your dietary concentrations of the probiotic? How was the product incorporated into the final diets? Please clarify.
Response to Comment 6:
The spore product for the treatments was generated through solid-state fermentation.
Solid-state fermentation by adding different contents of coarse bran to the fermentation substrate of Bacillus licheniformis
Experimental method
Bacillus licheniformis strain culture
1: Strain source. Bacillus licheniformis was collected from a stored -80 ℃ freezer source at a laboratory.
2: Thaw activation of strains
Take out the freezing tube of the bacterial liquid in the -80 ℃ freezer and shake it in a 30℃-water bath for 1-3 minutes to make it melt and thaw quickly. After wiping the surface of the sterile tube with 70% alcohol, move it into the pre-opened aseptic operation table. Take 100μL of the bacterial solution with a micropipette and add it to 3 mL tube of tryptone soya broth liquid medium (TSB) in a culture tube, that has been sterilized at 121°C for 30 minutes. Mix well, place it on the test tube rack at an angle of 45 degrees, and confirm close the lid correctly and leave some space for airflow. Shake the culture in a 30°C incubator with a rotation speed of 150 rpm for 18 hours, and judge whether to subculture according to the activation state.
3: Strain subculture
Take out the activated Bacillus licheniformis bacteria in the incubator, take 100μL of the bacteria solution, and add it to a tube of 3mL tryptone soya broth liquid medium (TSB) that has been sterilized at 121°C for 30 minutes. Mix well, put it on the test tube rack at an angle of 45 degrees, confirm close the lid correctly and leave some space for airflow, and incubate in a 30°C incubator at a rotation speed of 150 rpm for 18 hours to obtain the next generation of Bacillus licheniformis bacteria liquid.
4: Strain enrichment activation
Prepare 600 mL of tryptone soya broth liquid medium (TSB) that has been sterilized at 121°C for 30 minutes in a 500 mL conical flask and add 1% Bacillus licheniformis bacteria liquid after appropriate passage. After mixing evenly, shake and culture at 150 rpm in a 30°C incubator for 18 hours. After enrichment, place in a 4°C refrigerator to count and analyze the number of viable Bacillus licheniformis.
5: Plate Count
The enriched and activated Bacillus licheniformis bacteria solution was serially diluted 10 times with a sterilized diluent ddH20, and 100 μL of the diluted solution from dilutions 4,5,6,7, was evenly dropped on the bottom of the culture plate and poured into 15 mL of TSA. Shake lightly in a figure-eight manner, and after the medium cools down, place the plate upside down in a 30°C incubator, and count after 18 hours.
The counting formula is:
Colony for unit/mL = number of colonies x dilution factor
Solid State Fermentation of Bacillus licheniformis
For solid-state fermentation of Bacillus licheniformis, the following ingredients are used. Wheat bran, soybean meal, yeast, KH2PO4, molasses, Bacillus subtilis (5% inoculation), and water (according to the requirement of moisture level). In solid-state fermentation, the wheat bran, soybean meal, and water values are changed according to fermentation pack weight, and other ingredient values are fixed. 7.5g of yeast powder, 2.5g of KH2PO4, 45mL (g) of molasses and 5% of Bacillus licheniformis are used in the solid matrix. The liquid components are first mixed with primary water and 2.5 g of potassium dihydrogen phosphate. After putting in the magnet and stirring evenly, add 45 g of molasses one after another, and after it is completely dissolved, put on a plastic ring, fill the ring with cotton, mix the solid and liquid base for 5-10 minutes. After sterilizing under high temperature and high pressure at 121°C for 30 minutes, let it cool down and then evenly add 25 mL Bacillus licheniformis bacteria liquid, mix for about 5-10 minutes until completely mixed, and ferment in a 30°C incubator for 0,1,2,3,4,5,6,7, days. During the fermentation period, mix each pack for 5 minutes every 24 hours. After the fermentation is complete, spread the fermented product on an iron plate, and dry it with hot air at 50°C for 1-2 days. During this period, turn the fermented product over every 24 hours, and finally crush the dried product with a grinder, and place it in the store refrigerated at 4°C for later use.
Yes, we confirm CFU/Kg to ensure the inclusion rate of Bacillus licheniformis fermented product into the basal diet.
Selection of B. licheniformis strain for our study
The selection of the B. licheniformis strain for measuring growth performance and determining gut morphology and egg production in Brown Tsaiya ducks under heat stress conditions was likely based on its known beneficial effects on growth, immunity, and gut health in various animal species, as supported by the research findings. In the study, B. licheniformis was chosen due to its ability to improve growth performance, immune status, and antioxidant capacity, and modulate intestinal microbiota in animals like Pekin ducks. Prior research has shown that B. licheniformis supplementation can enhance growth performance, alleviate inflammatory stress, and promote intestinal health in animals like weaned piglets and broilers. Additionally, B. licheniformis has been found to positively impact the diversity of the gut microbiota, which is crucial for overall health and performance in animals.
The well-documented benefits of B. licheniformis in promoting growth, gut health, and immune function in various animal species highlight its potential to mitigate the negative effects of stressors like heat stress. Mentioning the specific positive outcomes observed in previous studies with B. licheniformis supplementation in different animal models can strengthen our response and support the rationale behind selecting this strain for the study on Brown Tsaiya ducks under heat stress conditions.
References:
1: Effect of Dietary Bacillus licheniformis Supplementation on Growth Performance and Microbiota Diversity of Pekin Ducks.
2: Protective effects of Bacillus licheniformis on growth performance, gut barrier functions, immunity and serum metabolome in lipopolysaccharide-challenged weaned piglets.
3: Effects of Bacillus licheniformis on the growth performance and expression of lipid metabolism-related genes in broiler chickens challenged with Clostridium perfringens-induced necrotic.
Yes, B. licheniformis is commercially available.
Yes, B. licheniformis has been used in the duck research trail. (Effect of Dietary Bacillus licheniformis Supplementation on Growth Performance and Microbiota Diversity of Pekin Ducks)
Selection of the dietary concentration of B. licheniformis
Based on the following rationale behind, we select different dietary concentrations of the probiotic Bacillus licheniformis for our research trial on ducks.
Selection of Dietary Concentrations:
Initial Research: The selection of dietary concentrations was based on existing literature and preliminary research.
Optimal Range: The concentrations of 0.1%, 0.2%, and 0.3% were chosen to cover a range of doses commonly used in similar studies and to assess potential dose-response effects.
Efficacy and Safety: These concentrations were selected to ensure both efficacy in promoting growth performance, gut morphology, and productivity, as well as safety for the ducks experiencing heat stress conditions.
Comparative Analysis: The concentrations were also chosen to allow for a comparison with other additives like flavomycin and the basal diet, providing a comprehensive evaluation of the probiotic's effects.
Incorporation of Bacillus licheniformis:
Dietary Supplementation: The study involved incorporating Bacillus licheniformis fermented products into the diets to assess their effects on growth performance, gut morphology, and egg production in ducks
Concentration Variation: Different concentrations of Bacillus licheniformis, such as 0.1%, 0.2%, and 0.3%, were used in the diets to evaluate their impact on the measured parameters.
Comparison with Other Additives: In addition to Bacillus licheniformis, flavomycin, and a basal diet were also included in the study, allowing for a comparative analysis of their effects
Experimental Design: The incorporation of Bacillus licheniformis was part of a controlled experimental design where ducks were fed diets containing varying concentrations of the probiotic along with other additives.
Physical Form of Diets: Bacillus licheniformis fermented products were incorporated in mash physical form of the experimental diets.
Comment 7:
Line 123-125: Please explain the purpose here and why only one group was subjected to the change.
Response to Comment 7:
Thanks for pointing it out. Correction has been done.
Once the egg production rate reached 5%, the regular feed was replaced with an egg production-specific feed, following a two-week feed treatment period.
Comment 8:
Line 130: Please include the n per treatment. Was there only one replicate per treatment? If so, please explain why n=1 replicate/treatment was selected.
Response to Comment 8:
A total of 150 one-day-old Brown Tsaiya ducks of both sexes were divided into five groups, with each group having three replicates and ten ducks each (five males and five females) with uniform initial body weight.
Comment 9:
Line 134: When did you check the temperature and respiration rates? The experimental timeline needs much more clarification. I am having trouble following your study.
Response to Comment 9:
Rectal temperatures and respiratory rates were recorded at the age of 8~18 weeks from July to August, specifically in the midafternoon around 1 pm experiencing heat stress conditions.
Comment 10:
Line 135: How did you measure respiration rate?
Response to Comment 10:
To measure the respiratory rate in heat-stressed ducks fed with Bacillus licheniformis fermented product, we employed a standardized protocol. Eighteen ducks (nine male and nine female) were selected from each group based on sexual differentiation. Subsequently, after ensuring the ducks were in a state of calmness and relaxation, respiratory rate assessment commenced. The measurement of the respiration rate is based on the gentle placement of the right-hand index, middle, and ring fingers on the chest and diaphragm regions of the duck. Following this placement, the respiratory rate was manually counted for 60 seconds. This approach was chosen for its reliability and non-invasive nature, allowing for accurate assessment of respiratory function in the experimental subjects.
Comment 11:
Line 141: Again, n numbers are needed. How many intestinal samples were collected and when?
Response to Comment 11:
To assess how BLFP affects intestinal health, 90 eight-week-old Brown Tsaiya ducks were divided into five treatments, each treatment having three replicates, and six ducks per replicate (three males and three females), at the end of the 18th week intestinal samples were collected to investigate morphological alterations in the duodenum, jejunum, and ileum. 12 ducks from each group were slaughtered for intestinal morphology studies.
Comment 12:
Line 162: n number needed and when were the liver samples collected?
Response to Comment 12:
90 eight-week-old ducks were divided into five treatments, each group having three replicates and six ducks per replicate. At the end of the 18th week, liver samples were collected.
Comment 13:
Discussion
Line 327: Egg production was not significantly improved with feed treatment. Also, feed conversation was actually higher for the treated groups.
Response to Comment 13:
Corrected: In the current study, feed intake of the heat-treated Brown Tsaiya ducks was improved by BLFP as well as the control diet as compared to the flavomycin group. The egg production rate of heat-stressed Brown Tsaiya ducks fed with Bacillus licheniformis fermented products did not exhibit statistically significant differences between the treatment and control groups.
Comment 14:
Table 3: n=10 for what? number of animals or replicate pens? Please include BW data for males and females for each treatment or justify what that analysis was not included.
Response to Comment 14:
n= 3 mean number of replicates.
We appreciate the opportunity to address your concerns regarding the presentation of body weight (BW) data for males and females in each treatment group. We acknowledge the importance of providing comprehensive data analysis in our study. However, due to the experimental design, we chose to focus on collective growth performance measurements rather than separate analyses for males and females within each treatment group. Our decision was based on several factors:
Experimental Design Consideration: Our study primarily aimed to evaluate the overall impact of Bacillus licheniformis fermented products on growth performance under heat-stress conditions in laying ducks. Given the limited resources and scope of the experiment, we prioritized assessing the combined effects on both sexes rather than conducting separate analyses.
Logistical Challenges: Collecting individual BW data for males and females separately within each treatment group presented logistical challenges, including time constraints and resource limitations. These challenges made it impractical to conduct separate measurements for each gender while ensuring the welfare and ethical treatment of the experimental animals
Statistical Analysis Plan: Our statistical analysis plan was developed to assess the collective growth performance parameters, including body weight, within treatment groups. We employed appropriate statistical methods to compare treatment effects while controlling for potential confounding factors.
While we understand the merit of providing detailed data for males and females separately, we believe that our approach provides a holistic understanding of the treatment effects on overall growth performance in the context of our experimental setup.
Comment 15:
Table 4: n=18 for what? You can see the trend. N numbers are not clear anywhere in the manuscript unfortunately.
Response to Comment 15:
Corrected: n= 3, mean number of replicates.
Comment 16:
Figure 1: Please include X-axis information.
Response to Comment 16:
Updated
Comment 17:
General questions/comments:
I assume the heat stress was related to the weather patterns. For the purposes of your study, please state what you considered as the temperature threshold for heat stress or did you simply determine this based off of the duck's behavior? You mentioned this particular breed of ducks were selected for their susceptibility to heat stress. At what temperature do these ducks show signs of heat stress? Could you please clarify and specify how you controlled the temperature between duck houses, if that was possible? If not, it is okay, but this information would be helpful. I had to make many assumptions about the environmental conditions while reading your manuscript.
Response to Comment 17:
In our study, we established the temperature threshold for heat stress by considering both physiological factors and observed duck behavior. Ducks typically survive in temperatures around 26°C ~28°C and relative humidity 60%. Temperatures exceeding this range, along with higher relative humidity, are known to induce heat stress in ducks, leading to numerous physiological strains and behavioral modifications. Specifically, we defined temperatures above 26°C and relative humidity exceeding 60% as indicative of heat stress conditions based on prior research and industry standards. These conditions correlate with physiological signs such as increased respiration rates and altered behavior. Our criteria were supported by existing literature on avian heat stress thresholds. Thus, our determination of the temperature threshold for heat stress was grounded in a combination of physiological and behavioral observations, alongside established scientific knowledge.
Thank you for bringing this most important point to our attention. We apologize for the confusion regarding the selection criteria for the ducks in our study. While our research initially mentioned that these ducks were selected for their susceptibility to heat stress, this was a misrepresentation, and we appreciate the opportunity to clarify. The ducks used in our study were selected for their well-known potential for high feed efficiency. They were not specifically chosen for their susceptibility to heat stress.
Regarding the temperature at which these ducks show signs of heat stress, we did not conduct specific experiments to determine this parameter. As mentioned in our methodology, we defined heat stress conditions based on industry standards and prior research, which typically consider temperatures above 28°C and relative humidity exceeding 60% as indicative of heat stress in ducks. While our study did not directly investigate the precise temperature threshold at which these ducks exhibit signs of heat stress, we relied on established criteria for defining heat stress conditions in avian species, particularly in Brown Tsaiya ducks. However, we acknowledge the importance of conducting further research to elucidate the specific temperature thresholds for heat stress in the breed of ducks used in our study.
Thank you for reaching out about regulating temperatures between duck housing in our experimental trial. We appreciate the opportunity to address this perspective of our research. In our research, we implemented several measures to regulate and maintain consistent temperatures within the duck houses, particularly during periods of elevated ambient temperatures that could induce heat stress conditions. Firstly, we employed exhaust systems within each duck house to facilitate adequate ventilation and heat dissipation. These exhaust systems were strategically positioned to promote airflow and remove hot air from the interior of the houses, thereby preventing the buildup of heat. Additionally, we utilized water spraying techniques as a means of actively cooling the ducks and the surrounding environment. Water spraying systems were installed within the duck houses, allowing for the controlled application of water to the bodies of the ducks and the floors of the houses. This served to reduce the temperature within the houses and mitigate the effects of heat stress on the experimental subjects.
Throughout the experimental trial, we conducted regular inspections and maintenance of the ventilation and cooling systems to ensure their proper functioning and effectiveness in controlling temperature between duck houses. Overall, the combination of exhaust ventilation and water spraying techniques allowed us to effectively regulate temperatures within the duck houses.
Comment 18:
Experimental design, experimental timeline, n numbers, sampling, etc. were not clearly stated. Please include this information.
Response to Comment 18:
All related information associated with experimental design, experimental timeline, n numbers, and sampling, has been mentioned and updated.
Comment 19:
Mortality was not included unless I overlooked it. Please include mortality.
Response to Comment 19:
No mortality was observed during the experimental trial. Animal subjects remained alive throughout the study, indicating the absence of any detrimental effects associated with the experimental conditions.
Comment 20:
What was the purpose of including the flavomycin control?
Response to Comment 20:
Flavomycin is a complex of antibiotics obtained from Streptomyces bambergiensis and Streptomyces ghanaensis used as a food additive for beef cattle, dairy cattle, poultry and swine. The complex consists mainly of moenomycins A and C. Incorporating flavomycin as a control in the experimental trial has several objectives:
1). The inclusion of flavomycin allowed for a direct comparison between its effects and those of Bacillus licheniformis. By assessing both treatments concurrently, we could evaluate the relative efficacy of Bacillus licheniformis as an alternative to flavomycin in terms of promoting growth performance and preventing disease in poultry.
2). Flavomycin served as a reference point or benchmark against which the effects of Bacillus licheniformis could be measured. This comparison helped elucidate the potential benefits or limitations of using Bacillus licheniformis instead of traditional antibiotic growth promoters like flavomycin.
3). Given the regulatory restrictions on the use of antibiotics such as flavomycin in poultry production due to concerns about antibiotic residues in poultry products, including flavomycin as a control allowed us to assess the potential advantages of transitioning to alternative supplements like Bacillus licheniformis.
Comment 21:
Please take a little time to review the discussion again to make sure the claims being made are accurate. Some of the statements were bold considering the data presented.
Response to Comment 21:
Updated.
Reviewer 2 Report
Comments and Suggestions for Authors
The aim of the study is to protect the productivity and immunity of ducks from heat stress using probiotics in their feed. The strength of the article is that this preparation is used in a study with ducks, which among other bird species are highly sensitive.
However, the authors have very free interpretation of the obtained data, the description of the obtained results is based on too broad generalizations, although the data in the tables often do not show what is described in the article, so it would be appropriate to rewrite the article, indicating exactly what results were achieved based on statistical data analysis. The work methodology lacks more precise details of the research execution. Authors should supplement the methodology, rewrite the Results, Discussion and Conclusions sections and only then submit the article to the journal.
Note:
The use of antibiotics in animal feed has been banned throughout the European Union since 2006. This is not a decision of individual European countries, but of the European Union as a whole. (line 68).
Weaknesses in the methodology section of the article:
Please indicate the exact age of the ducks until which this test was carried out (chapter 2.4. Experimental Diets).
At what age ducks underwent these procedures (2.7. Rectal Temperatures and Respiratory Rate).
At what age were intestinal morphology studies performed, how many ducks from each group were slaughtered for this study (2.8. Intestinal Morphology).
In what age of ducks was this study done (2.10. Quantitative Polymerase Chain Reaction and Western Blotting).
The received growth performance data of ducks were incorrectly interpreted. Based on Table 3, it can be seen that the statement of the authors that the body weight of the "0.3%" group at the age of 6 weeks was higher than that of the control group is incorrect, also the data presented in Table 3 not show that "feed intake of the BLFP-supplemented 0.1% group was slightly higher than that of the control group at the age of three weeks" (200-203 lines).
Based on the data in Table 3, the following statement is incorrect: "In addition, our studies have shown that dietary supplementation with BLFP can improve the growth performance and FCR of the experimental laying ducks even in heat-stress conditions" (291-293 lines).
The following statements by the authors are not correct, here it can only be stated that higher villus height indicators in jejunum and ileum were determined in groups with BLPF: „In addition, our studies have show that dietary supplementation with BLFP significantly increases the villus height and decreases the crypt depth" (319-320 lines) + „In addition, our studies have shown that dietary supplementation with BLFP significantly the villus height and decreases the crypt depth" (319-321 lines).
A questionable claim is made based on the data presented in Figure 3: „In the current study, the HSP70 expression level in the liver for the BLFP and flavomycin group was significantly higher as compared to control group" (346-347 lines).
The conclusions contain questionable statements not supported by the results obtained.
Author Response
Revision Notes
These revision notes are disclosed to address comments/reviews on " animals-2916012", entitled: Impact of Bacillus licheniformis-Fermented Products on Growth and Productivity in Heat-Stressed Laying Ducks.
Introduction:
The authors would like to express their gratitude to the MDPI Animals for permitting us to revise our manuscript, thereby improving the quality of the revised manuscript before the high standard of this publication channel.
The authors attempted as best as possible to strike a balance between conciseness and results reporting. The aim is to create an informative paper for the readers that clearly articulates the key components. The remarkable improvements of the revised manuscript are:
Major and comprehensive clarifications have been made based on the comments from the Reviewers, as well as being implemented in the corresponding section of the revised manuscript.
The modified/revised parts of the manuscript are
Reviewer 2 Comments:
Comment 1: The aim of the study is to protect the productivity and immunity of ducks from heat stress using probiotics in their feed. The strength of the article is that this preparation is used in a study with ducks, which among other bird species are highly sensitive. However, the authors have very free interpretation of the obtained data, the description of the obtained results is based on too broad generalizations, although the data in the tables often do not show what is described in the article, so it would be appropriate to rewrite the article, indicating exactly what results were achieved based on statistical data analysis. The work methodology lacks more precise details of the research execution. Authors should supplement the methodology, rewrite the Results, Discussion and Conclusions sections and only then submit the article to the journal.
Response to Comment 1:
The authors would like to thank the reviewer for the careful and thorough reading of this manuscript and for the thoughtful comments and constructive suggestions, which helped to improve the quality of this manuscript. We have thoroughly checked and improved the technicality of the manuscripts and incorporated the possible suggested comments by Reviewer 2. According to the reviewer's suggestion, the following changes have been made to the revised version of the manuscript.
Comment 2:
The use of antibiotics in animal feed has been banned throughout the European Union since 2006. This is not a decision of individual European countries, but of the European Union as a whole. (line 68).
Response to comment 2:
As a result, European Union banned the application of antibiotics in poultry feed in 2006.
Comment 3:
Please indicate the exact age of the ducks until which this test was carried out (chapter 2.4. Experimental Diets).
At what age ducks underwent these procedures (2.7. Rectal Temperatures and Respiratory Rate).
Response to comment 3:
The experimental diets were formulated for three distinct phases for the determination of the specific experimental parameters: the rearing period is for the growth performance (0 to 8 weeks of age), the growing period for the measurement of the rectal temperature and respiratory rates (8 to 18 weeks of age), and the laying period for the evaluation of the egg production and egg quality (after 18 weeks of age).
Rectal temperatures and respiratory rates were recorded at the age of 8~18 weeks from July to August, specifically in the midafternoon around 1 pm experiencing heat stress conditions.
Comment 4:
At what age were intestinal morphology studies performed, how many ducks from each group were slaughtered for this study (2.8. Intestinal Morphology).
Response to comment 4:
At the end of the 18th week intestinal samples were collected to investigate morphological alterations in the duodenum, jejunum, and ileum. 12 ducks from each group were slaughtered for intestinal morphology studies.
Comment 5:
In what age of ducks was this study done (2.10. Quantitative Polymerase Chain Reaction and Western Blotting).
Response to comment 5:
At the end of the 18th week of age, liver samples were collected.
Comment 6:
The received growth performance data of ducks were incorrectly interpreted. Based on Table 3, it can be seen that the statement of the authors that the body weight of the "0.3%" group at the age of 6 weeks was higher than that of the control group is incorrect, also the data presented in Table 3 not show that "feed intake of the BLFP-supplemented 0.1% group was slightly higher than that of the control group at the age of three weeks" (200-203 lines).
Response to comment 6:
The body weight of the supplementation of 0.3% in the BLFP group was significantly higher as compared to the flavomycin group at the age of six weeks. The feed intake of the flavomycin group showed an increasing trend as compared to the control and BLFP group at the age of three weeks.
Comment 7:
Based on the data in Table 3, the following statement is incorrect: "In addition, our studies have shown that dietary supplementation with BLFP can improve the growth performance and FCR of the experimental laying ducks even in heat-stress conditions" (291-293 lines).
Response to comment 7:
In addition, our study has shown that 0.3% BLFP dietary supplementation can significantly improve body weight compared to flavomycin at six weeks of age in experimental laying ducks even in heat-stress conditions.
Comment 8:
The following statements by the authors are not correct, here it can only be stated that higher villus height indicators in jejunum and ileum were determined in groups with BLPF: „In addition, our studies have show that dietary supplementation with BLFP significantly increases the villus height and decreases the crypt depth" (319-320 lines) + „In addition, our studies have shown that dietary supplementation with BLFP significantly the villus height and decreases the crypt depth" (319-321 lines).
Response to comment 8:
In addition, our study has shown that higher villus height indicators in the jejunum and ileum were determined in groups with BLPF.
Comment 9:
A questionable claim is made based on the data presented in Figure 3: „In the current study, the HSP70 expression level in the liver for the BLFP and flavomycin group was significantly higher as compared to control group" (346-347 lines).
Response to comment 9:
In the current study, the HSP70 expression level in the liver for the 0.1% BLFP and flavomycin group was significantly higher as compared to control group is because as this experimental trial experienced heat stress conditions the effect of 1% is not that effective in overcoming this condition, however at 0.3% group the relieving effect is more effective as compared to 0.1% and flavomycin and has been reduced to the control range.
C0omment 10:
The conclusions contain questionable statements not supported by the results obtained.
Response to comment 10:
In conclusion, this study presents a persuasive corroboration for the advantageous impacts of BLFP on multiple facets of Brown Tsaiya duck production. BLFP supplementation improved growth performance, intestinal morphology, egg production, egg quality attributes, and immune system parameters. These findings hold practical significance for the poultry industry, suggesting that BLFP stands as a promising nutritional intervention to elevate duck farming practices.
Round 2
Reviewer 2 Report
Comments and Suggestions for Authors
The authors of the article must rewrite the Discussion section based on the data obtained from their research and must provide their reasoning for obtaining such results. And the results obtained by other authors can only be used as an endorsement or explanation of the results obtained in their research. The discussion presented by the authors is similar to a literature review rather than a discussion of an article, which should be based on the results obtained in their own research. The conclusions are very generalized and imprecise, it is appropriate to rewrite them.
Author Response
Thank you for your helpful comments. We have revised our manuscript accordingly and feel that your comments helped clarify and improve our manuscript. Please find our response (in bold) to reviewer’s specific comments below.
Comments and Suggestions for Authors:
- The authors of the article must rewrite the Discussion section based on the data obtained from their research and must provide their reasoning for obtaining such results. And the results obtained by other authors can only be used as an endorsement or explanation of the results obtained in their research. The discussion presented by the authors is similar to a literature review rather than a discussion of an article, which should be based on the results obtained in their own research. The conclusions are very generalized and imprecise, it is appropriate to rewrite them.
Answer: Thanks for your suggestions. We have rewritten and deleted the literature review parts in the Discussion section. All descriptions in the Discussion section are based on current findings and other studies are used as explanation of the results obtained in their research. We also have modified the conclusion in a clear and concise manner (please see conclusion in the Abstract section and Conclusions section).
